# Variables appended to ABS frames: Has their data quality improved?

**Shelley Roth**[�], **Andrew Caporaso**[ID]*[☬], **Jill DeMatteis**[☬]

Westat, Rockville, Maryland, United States of America

☬ These authors contributed equally to this work.
* andrewcaporaso@westat.com

## Abstract

Address based sampling (ABS) has become current state-of-the-art methodology for conducting household surveys by mail, telephone or web in the United States. One potential advantage of ABS frames is that additional information about the sampled households can be appended and leveraged for data collection and analytic purposes. The appended data come from many sources and are of variable quality and completeness. The goals of this research were to evaluate data quality of demographic and socioeconomic variables provided for recent ABS samples from one vendor, and to examine their potential usefulness for sample design, including oversampling. We report on the completeness of the appended data as well as their concordance with data reported by respondents to two recent large ABS household surveys, one that invited households to complete the survey online and another that was mail only. Based on the quality assessment, we also examine the utility of the appended variables for oversampling. Our general conclusions are that the quality of select appended variables has improved such that the Hispanic origin, Hispanic surname, and presence of age group 65+ variables may be used to efficiently oversample these subgroups. However, this is not the case for oversampling other subgroups through appended variables for home tenure; those with head of household whose educational attainment is high school or less; low income households; households with children; presence of age groups 18–24, 25–34, and 35–64; or households based on the number of adults in the household.

## Introduction

Address based sampling (ABS) has become current state-of-the-art methodology for household surveys in the U.S. ABS is based on residential address frames largely derived from U.S. Postal Service (USPS) files. The ABS frames maintained by reputable vendors provide nearly complete coverage of residential mailing addresses in the U.S. [1, 2]. Furthermore, the ability to mail to the addresses, to invite participation in a web survey, to visit the addresses for in-person interviewing, or to append landline telephone numbers to the addresses offers a variety of choices for contact and data collection methods.

In addition to telephone numbers, vendors are able to append other characteristics of the address that can be leveraged for data collection (for example, number of household members

**Data Availability Statement:** We are unable to share a dataset containing the vendor-appended variables because these data are owned by a third-party organization (specifically, MSG, the vendor we used for these samples). The data sharing restrictions are imposed by the contract ("Master

Services Agreement") Westat holds with MSG. The restrictions on sharing respondents' data are imposed by the Confidential Information Protection and Statistical Efficiency Act (CIPSEA) pledge that was provided to our survey respondents. Data requests for vendor-appended data may be made by contacting MSG (www.m-s-g.com). Note that MSG is not at liberty to provide the vendor-appended data for the NHTS, NHES, and HINTS samples used in this research. For the NHTS data, please see: https://nhts.ornl.gov/downloads. For the HINTS data, please see: https://hints.cancer.gov/data/default.aspx. While users can replicate our methods, they cannot access the NHES data to validate our specific results due to third-party data sharing restrictions. The National Center for Education Statistics owns the NHES survey data and our appended variable vendor MSG owns the appended frame data. We are not authorized to release either the NHES or MSG data. The restrictions are in place to protect the confidentiality of the NHES respondents. Please reach out to the following contacts with questions about the data sources used in this report. NHES: Andrew Zukerberg, Senior Research Scientist, Sample Surveys Division, NCES (Andrew.Zukerberg@ed.gov) NHTS: Danny Jenkins, NHTS Program Manager, FHWA (Daniel.jenkins@dot.gov) HINTS: Kelly D. Blake, Director, Health Information National Trends Survey (HINTS): kelly.blake@nih.gov.

**Funding:** All authors are employees of Westat. There are no patents, products in development or marketed products to declare. This does not alter our adherence to PLOS ONE policies on sharing data and materials.

**Competing interests:** The authors have declared that no competing interests exist.

or whether the household members rent or own the house). This appended data has been used or evaluated for a number of data collection purposes, including predicting household eligibility [3], stratification and sample allocation [4–6], and other purposes such as enhancing efficiency of survey operations [7, 8]. [4] found some promise only in Hispanic origin or surname, but otherwise found availability and quality of appended variables needed improvement before considering them for sample design. [6] also found potential value in using Hispanic origin or surname, as well as presence of age 65+ in the household, in improving efficiency of locating members of those domains. [8] found potential in home tenure, age, marital status and gender, and found race/ethnicity to be of mixed value in predicting eligibility.

These studies and others [9] have concluded that the quality of the appended ABS data are major barriers for leveraging the data successfully. In 2016, [10] concluded that leveraging vendor-appended frame data should be done "with a healthy dose of skepticism" (section 3.2.2).

[10] discusses a number of reasons why appended frame data should be used with caution. Such data are often proprietary so frame vendors are reluctant to discuss how information was collected. The data are typically appended for commercial purposes that may not meet accuracy and completeness standards for scientific research. The data come from numerous sources that can vary in completeness, timeliness, and accuracy. Data about one individual household member (e.g. marital status) may be applied to the entire household.

Vendors of commercial lists are continuously enhancing their data sources and methodology for appending new data [9]. Thus, it is important to continue to periodically reassess the quality of the appended data and its viability for use in improving sample efficiency and other purposes. This research is in part a follow-up to [4] to determine whether the accuracy and completeness of appended frame data have changed over time. High quality frame information can be used in survey design to gain efficiencies in sampling and estimating outcomes in hard-to-reach populations. We selected fourteen items for evaluation based on their availability on the vendor's frame and in survey questionnaires. Here, we evaluate the quality of the demographic and socioeconomic items that were appended to two ABS samples and examine the feasibility of using these appended demographics for stratification or oversampling.

## Data and methods

This research looks at appended data from one sample vendor; the sources, methods, and quality of these appended variables are likely to vary by vendor. All of the variables we examine are appended by the vendor through address matching, and might not correspond to the current residents or a particular resident of the address. The fourteen variables we evaluated are listed below. Those variables with a * are the ones that were only available in the HINTS data:

- Head of household (HH) is of Hispanic origin

- Head of HH has Hispanic surname

- Home is rented

- Head of HH Education = < HS

- Head of HH Income <$35K

- Presence of children in HH

- Adult age 18–24 present in HH*

- Adult age 25–34 present in HH*

- Adult age 35–64 present in HH*

- Adult age 65+ present in HH*

- 1 adult HH*

- 2 adult HH*

- 2+ adult HH*

- 3+ adult HH*

We evaluate the quality of appended variables by comparing these data to information reported by respondents from two surveys conducted in 2017: the National Household Travel Survey (NHTS) and the Health Information National Trends Survey 5, Cycle 1 (HINTS). To evaluate change over time, we compare the findings to similar research conducted by [4], who performed a similar comparison of appended data and respondent reports from the 2011 Field Test of the National Household Education Survey (NHES).

To evaluate using the appended variables for stratification, we examine the effect associated with oversampling targeted subgroups (based on appended frame data) on nominal and effective yield for the subgroup and on overall effective yield. Because the NHTS has a much larger sample size than HINTS, where possible, we used the NHTS survey and frame data for the evaluations of availability and quality. Findings using HINTS data were very similar to the findings based on the NHTS data for variables that overlapped between the two studies. For variables where NHTS data were not available, we used the HINTS data. The 2011 NHES data are used in Fig 1 to illustrate change in appended variable availability since then.

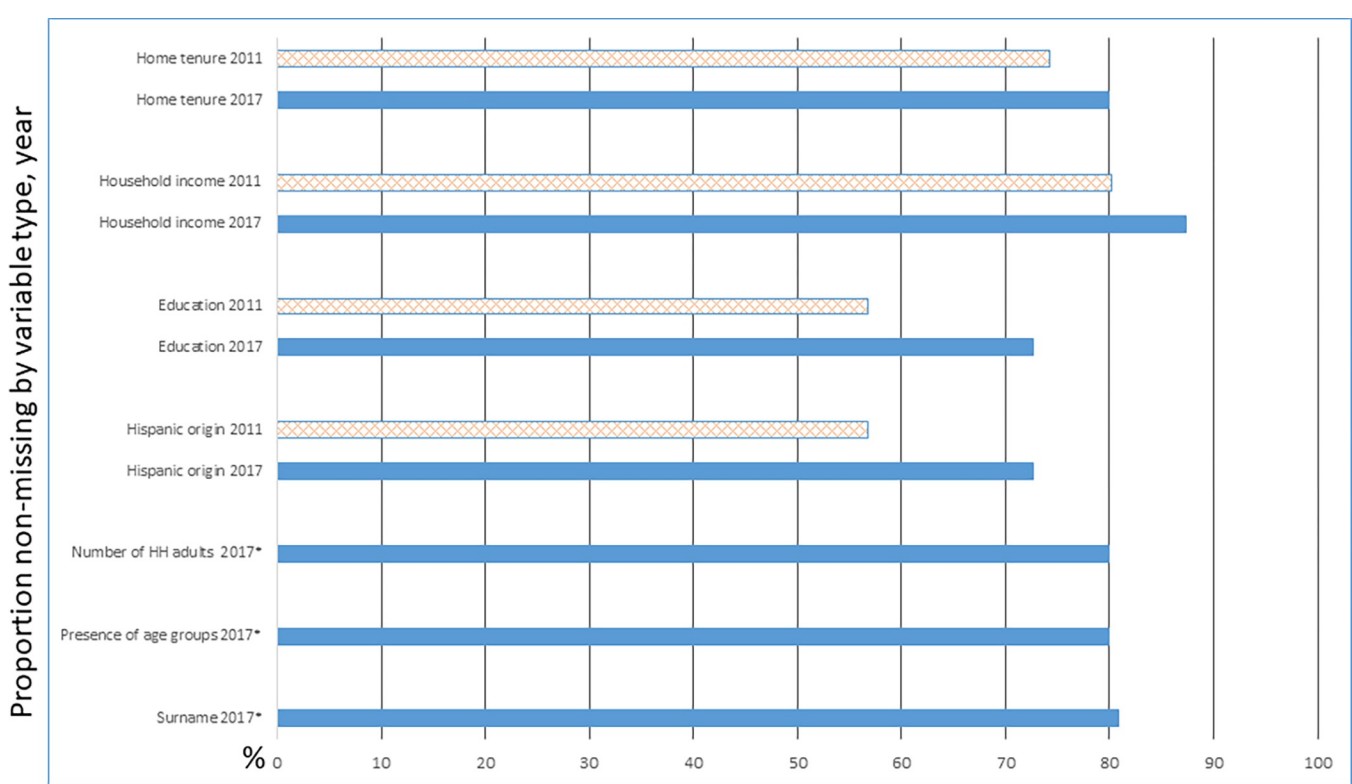

**Key: 2011 (hashed bars); 2017 (solid bars)**
*Did not examine these characteristics in evaluation based on 2011 study.

**Fig 1. Non-missing rate for appended demographics.**

**Table 1. Household studies used for analyses.**

| Study name | Study description | Study sponsor | Geographic area | Target population | Number of households sampled[1] | Number of completed surveys[1] | Screener response rate (AAPOR RR3) | Extended survey conditional response rate (AAPOR RR2) |
|---|---|---|---|---|---|---|---|---|
| **2017 National Household Travel Survey (NHTS)[2]** | *Two-phase national household ABS with oversampling in some geographic areas* | US DOT Federal Highway Administration | US 50 states + DC | All individuals 5 years old and up in households | 929,000 | 130,000 | 30% | 52% |
| **2017 Health Information National Trends Survey (HINTS)[3a3]** | *Single-phase national household ABS with oversampling of high minority areas* | National Cancer Institute | US 50 states + DC | Households with at least 1 adult | 13,300 | 3,300 | NA | 32% |
| **2011 National Household Education Survey Field Test (NHES)[4]** | *Two-phase national household ABS* | National Center for Education Statistics | US 50 states + DC | Households with children | 41,260 | 5,590 | 69% | 73% |

[1] These figures are rounded counts.

[2] See https://nhts.ornl.gov/assets/DE_public.pdf for details on the NHTS design and methodology.

[3] See https://hints.cancer.gov/ for details on the HINTS design and methodology.

[4] See [11] for details on the NHES design and methodology

Table 1 contains details of the three studies we used for the analyses. See the S1 Appendix for the exact wording of the survey questions used in these analyses.

## Availability and quality of appended demographic information

The appended demographic information we assessed is listed in Table 2. The items included: Hispanic origin of the head of household, Hispanic surname flag, presence of someone 18–24, 25–34, 35–64, or 65+ in the household, home tenure, presence of children in the household,

**Table 2. Predictivity and concordance results.**

| Characteristic | Positive predictivity | | Overall concordance |
|---|---|---|---|
| | (True +) | (True—) | |
| Hispanic origin[1] | 0.72 | 0.93 | 0.91 |
| Hispanic surname[1] | 0.79 | 0.90 | 0.90 |
| 18–24 present[2] | 0.44 | 0.92 | 0.88 |
| 25–34 present[2] | 0.32 | 0.88 | 0.80 |
| 65+ present[2] | 0.78 | 0.81 | 0.80 |
| Home is rented[1] | 0.83 | 0.77 | 0.79 |
| Presence of children[1] | 0.76 | 0.58 | 0.73 |
| 3+ adult HH[2] | 0.27 | 0.87 | 0.72 |
| Head of HH Education = < HS [1] | 0.37 | 0.84 | 0.71 |
| Income <$35K[1] | 0.66 | 0.73 | 0.70 |
| 35–64 present[2] | 0.80 | 0.60 | 0.70 |
| 1 adult HH[2] | 0.46 | 0.77 | 0.67 |
| 2+ adult HH[2] | 0.81 | 0.46 | 0.65 |
| 2 adult HH[2] | 0.64 | 0.52 | 0.55 |

[1]Source: 2017 NHTS

[2]Source: 2017 HINTS

the household has 1 adult, 2 adults, 2+ adults, or 3+ adults, the householder has a high school education or less, and household income is less than $35,000. Non-missing rates, computed as the proportion of sampled addresses for which the particular appended variable was available, are shown in Fig 1. The non-missing rates for presence of adults in certain age groups (e.g., 18–24 year old present) are the same for all of the age groups reported in Table 1. The same is true for total number of household adults and different income levels. Results from the 2017 surveys indicate that there has been improvement in availability of the appended data since 2011 for all of the variables that overlap between 2017 and 2011. However no appended variable is available for more than 90 percent of the sampled households.

Next, we examined the accuracy of the appended data by comparing it to survey responses. All data were categorized into dichotomous "yes" and "no or missing" groups indicating presence of each characteristic; for example, presence of children at each address was dichotomized to "yes" and "no or missing". In Table 2 we present agreement statistics between frame data and survey responses for each appended variable, including true positive, true negative, and overall concordance rates [12]. The true positive rate is the percent of responding households with the characteristic of interest (based on respondent reports) for which the appended variable also indicates presence of the characteristic. The true negative rate is calculated similarly for the absence of a characteristic. The overall concordance rate is the proportion of cases for which the appended value matches the respondent reported value. For example, for the first row of Table 2, Table 3 shows a cross-tabulation of the survey data vs. the appended data showing weighted household counts for each of the cells:

This shows that an estimated 72% of responding households who reported being of Hispanic origin on the NHTS survey were flagged as such by the appended frame data (= 7,270,024/10,135,167), and an estimated 93% of households who reported not being Hispanic were not flagged as Hispanic in the frame data (= 100,196,096/107,963,499). Overall, an estimated 91% of responding households' survey responses were in agreement with the frame data (= (7,270,027 + 100,196,096) / 118,098,666).

Most notably, both the true negative rate and overall concordance rate are notably high for several items, including Hispanic origin, Hispanic surname, and presence of someone age 18–24 in the household. The highest true positive rates were observed in Hispanic surname, home tenure, 2+ adult households, and presence of age groups 35–64 and 65+.

## Efficiency of using appended demographic variables for oversampling

Many surveys assess outcomes for hard-to-reach population subgroups. Oversampling these groups can help with achieving a desired level of precision for outcome estimates. Using appended demographic variables is potentially an efficient way to oversample hard-to reach populations, so we analyzed the fourteen variables shown in Table 2 to determine their potential use for oversampling. For each characteristic, the same two categories indicating presence of the characteristic ("yes" and "no or missing") were used to define strata. For each variable,

**Table 3. Example of weighted household counts used to compute concordance of ethnicity data between vendor-appended frame data and survey data.**

| Survey data | Vendor-appended data | | |
|---|---|---|---|
| | **Flagged as Hispanic** | **Not flagged as Hispanic** | **Total** |
| Hispanic | 7,270,024 | 7,767,403 | 15,037,427 |
| Not Hispanic | 2,865,143 | 100,196,096 | 103,061,239 |
| Total | 10,135,167 | 107,963,499 | 118,098,666 |

we examined several scenarios involving different oversampling rates for the stratum containing the subgroup (based on the appended variable).

We used three measures to quantify the effects of oversampling relative to an equal probability selection mechanism: the nominal increase in yield for the subgroup, the change in effective yield for the subgroup (which adjusts the nominal increase in yield to account for the design effect (DEFF) due to differential sampling as well as misclassification), and the design effect for overall (i.e., not subgroup) estimates. See [13], section 11.7C, for design effect formulae.

Let $n_{h0}$ = the sample size for subgroup h under an equal probability selection mechanism;

$n_{hk}$ = the sample size for subgroup h when the high-density stratum is oversampled by a factor of k;

$deff_{hk}$ = the expected design effect (accounting for both differential sampling and misclassification) for subgroup h when the high-density stratum is oversampled by a factor of k; and

$deff_{\cdot k}$ = the expected overall design effect (accounting for both differential sampling and misclassification) when the high-density stratum is oversampled by a factor of k.

Then the nominal increase in yield for subgroup h when the high-density stratum is oversampled by a factor of k is the ratio $n_{hk}/n_{h0}$ and the change in effective yield for the subgroup is the ratio $\left(\frac{n_{hk}}{deff_{hk}}\right)/n_{h0}$.

For two variables (one well-performing and one not well-performing, as defined by the effect of oversampling on the effective yield identified via our analysis), Fig 2 plots the three measures for relative sampling rates ranging from 1 to 3 for the targeted subgroup. S1 Fig presents the plots for all of the variables under evaluation, sorted by the overall concordance rates presented in Table 2.

Fig 2 and S1 Fig show plots with three curves with the x-axis showing the relative oversampling rate in the high density stratum (k). The solid line shows the nominal increase in yield for the subgroup of interest, the dotted curve shows the effect of oversampling on the effective yield, and the dashed curve shows the overall design effect due to oversampling and misclassification.

For each characteristic, the nominal yield for the subgroup increases linearly as the oversampling rate increases. That is, as expected, subgroup yield may be increased by oversampling the targeted subgroup. However, these increases are nominal increases; the more disproportionate the sampling rates due to oversampling of the targeted subgroup, the greater the impact

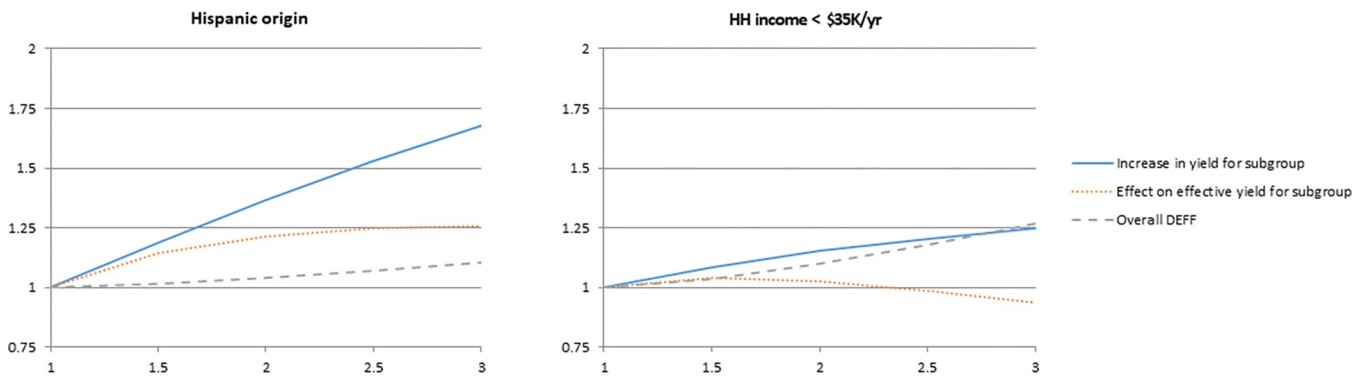

Note: The x- axis shows the oversampling ratio and the y-axis shows the change in yield, effective yield, and DEFF (see legend)

**Fig 2. Effect of oversampling on yield, effective yield, and overall design effect (DEFF) for two appended demographic characteristics from ABS frame.**

on (i.e., reduction in) effective yield, first for general population estimates and subgroup estimates. These patterns are illustrated by the other curves in each plot in Fig 2 and S1 Fig. The dotted line, showing the impact of oversampling on the effective yield for the subgroup, generally rises to a point, and then declines. The overall DEFF (dashed line) increases monotonically.

In practical terms, the decision of whether to oversample based on a particular characteristic and the rate at which to oversample requires balancing the gains in effective sample size for the subgroup against the declines in the effective sample size for overall estimates.

Based on this analysis, we used a general rule of an oversampling scenario where k ≥ 1.5 and the increase in effective yield for the subgroup is greater than 10 percent while the overall design effect remains below 1.1, to identify which of the variables have potential for use in oversampling. These include Hispanic origin (shown in Fig 2), Hispanic surname, and presence of age group 65+. For each of these promising variables, we computed an optimum oversampling rate, shown in the three shaded rows of Table 4. This rate is computed as $\sqrt{p_H / p_L}$, where $p_H = \hat{M}_H / \hat{N}_H$ is, for the high-density stratum H, the estimated prevalence of the rare population $M_H$ in the total population $N_H$, and $p_L = \hat{M}_L / \hat{N}_L$ is, for the low-density stratum L, the estimated prevalence of the rare population $M_L$ in the total population $N_L$ [14]. For example, using weighted household counts for Hispanic origin, we illustrate the calculations for the first row in Table 4 as follows: $\hat{M}_H = 7,270,024$, $\hat{N}_H = 10,135,167$, $\hat{M}_L = 7,767,403$, and $\hat{N}_L = 107,963,499$, so $p_H = \frac{7,270,024}{10,135,167} = 0.72$, $p_L = \frac{7,767,403}{107,963,499} = 0.07$, an the optimum oversampling rate is $\sqrt{0.72 / 0.07}, = 3.2$. While these are optimum for the particular subgroup, for a particular survey these oversampling rates would likely be scaled back to mitigate the effects of the differential sampling on the precision of estimates for other subgroups of interest.

We also computed optimum oversampling rates for the less promising variables. As described above, the oversampling appears to be worthwhile for the three shaded variables in Table 4 since there is a high prevalence of the characteristic in the high-density stratum and a low prevalence of the characteristic in the low-density stratum. However, for the other

**Table 4. Optimum oversampling rates.**

| Characteristic | $p_H$ | $p_L$ | Optimum oversampling rate |
|---|---|---|---|
| Hispanic origin[1] | 0.72 | 0.07 | 3.2 |
| Hispanic surname[1] | 0.79 | 0.10 | 2.8 |
| 18–24 present[2] | 0.44 | 0.08 | 2.3 |
| 65+ present[2] | 0.78 | 0.19 | 2.0 |
| Home is rented[1] | 0.83 | 0.23 | 1.9 |
| Income <$35K[1] | 0.66 | 0.27 | 1.6 |
| 25–34 present[2] | 0.32 | 0.12 | 1.6 |
| Education HS or less[1] | 0.37 | 0.27 | 1.5 |
| 3+ adult HH[2] | 0.27 | 0.13 | 1.5 |
| 35–64 present[2] | 0.80 | 0.40 | 1.4 |
| 1 adult HH[2] | 0.46 | 0.23 | 1.4 |
| Presence of children[1] | 0.76 | 0.42 | 1.3 |
| 2 adult HH[2] | 0.64 | 0.48 | 1.2 |
| 2+ adult HH[2] | 0.81 | 0.54 | 1.2 |

[1]Source: 2017 NHTS
[2]Source: 2017 HINTS

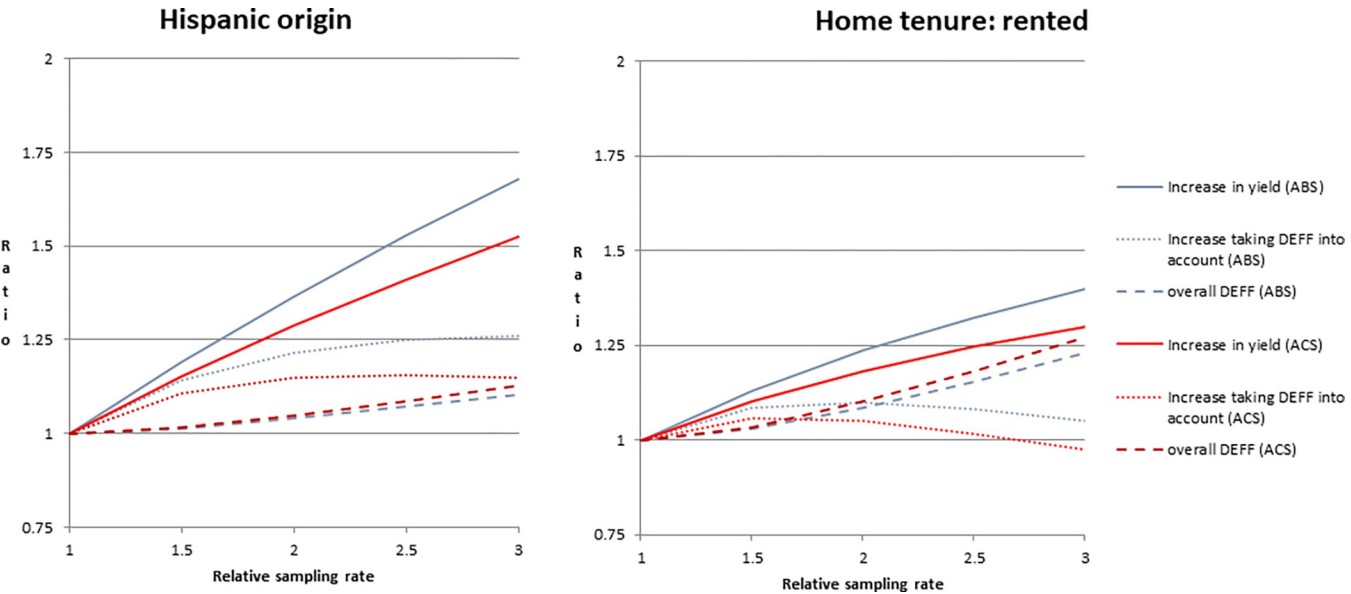

**Fig 3. Effect of oversampling on yield, effective yield, and overall design effect (DEFF) for Hispanic origin and home tenure based on aggregated census tract-level data from the American Community Survey (ACS) in red and appended frame data (ABS) in blue.**

characteristics, the oversampling is not as effective because the difference between the prevalence in the high-density stratum and low-density stratum is not as large. Fig 2 clearly illustrates the difference that oversampling can make in effective yield, showing one of the promising variables, Hispanic origin, next to one of the less promising variables, low income households.

While the appended variables offer a relatively new option for oversampling, another alternative method that has long been used for household surveys is to oversample based on aggregate (e.g., census tract-level) characteristics [15, 16]. For Hispanic origin and home tenure, we compared oversampling using appended variables from a vendor to oversampling using aggregate-level data from the American Community Survey (ACS). We repeated the analysis described above and found similar results when using the 5-year (2012–2016) ACS tract-level characteristics to oversample. (See Fig 3 for comparison of aggregated ACS data in red and appended frame data in blue). For both variables, the appended data appears to slightly outperform the aggregated data in terms of increase in effective yield as the relative sampling rate increases.

## Discussion

In this study, we examined the availability and quality of variables appended to the ABS frame and discussed the viability of using the variables to improve sample designs. Our general conclusion in this research is that availability of the appended variables has improved over the past decade, and that the quality of some of the variables we considered is sufficient for use in oversampling. For other variables, data quality is not sufficient for this purpose. Specifically, we concluded that oversampling target subgroups defined by Hispanic origin, Hispanic surname, and presence of age group 65+ may result in modest increases in subgroup effective yield with minimal effect on overall effective yield; the finding that these are the most promising appended variables is consistent with the findings reported by ([4, 6, 8]), but we also have the finding that these variables are now of sufficient quality to be effective for oversampling. This

is not the case for oversampling other subgroups, including home tenure, those with head of household whose educational attainment is high school or less, low income households, households with children, presence of age groups 18–24, 25–34, and 35–64, 1 adult households, 2 adult households, 2+ adult households, or 3+ adult households. These findings are also consistent with those reported in the earlier studies ([4, 6, 8]).

For two variables, Hispanic origin and home tenure, a comparison of the effectiveness for oversampling using the appended variables to the effectiveness of oversampling using tract-level characteristics from the ACS revealed similar results, with the appended variables slightly outperforming the ACS tract-level characteristics. A possible extension of this work would be to consider using the appended variables (e.g., appended Hispanic origin) in conjunction with the tract-level characteristics to stratify and oversample; for example, the "high-density Hispanic" stratum would include addresses that either are flagged as Hispanic based on the appended indicator or are located in tracts with relatively high proportions of Hispanics.

One limitation of this study is that validation information is only available from survey respondents, which limits the variables that could be examined, and the number of households whose appended frame data could be validated. Additionally, the respondent-provided information is treated as the "truth" for the household; any measurement error in these responses is not accounted for in this analysis. We also only looked at a subset and not all of the available appended variables that could be considered for use in oversampling.

Finally, the findings pertain to the variables appended by one particular vendor; for other vendors, similar evaluations could be undertaken to determine whether the appended data are of sufficient quality to use for oversampling. The quality of the appended data is influenced by how these appended variables are created; for example, the use of credit or market research data will tend to underrepresent household with lower socioeconomic status. The methods for appending frame data are often unavailable to researchers and the data are not necessary provided for research purposes.

Future research should continue to monitor and document the availability and accuracy of different types of appended frame data. Since 2013 [4], quality improved in the presence of age group 65+ variable enough to consider using it for oversampling. As the quality continues to improve, so too will the ability to successfully leverage additional appended variables for sampling and other research purposes.

## Supporting information

**S1 Fig. Appendix.** Note: The horizontal axis shows the oversampling ratio.
(TIF)

**S1 Appendix. Variables appended to ABS frames.**
(DOCX)

## Author Contributions

**Conceptualization:** Shelley Roth, Andrew Caporaso, Jill DeMatteis.

**Data curation:** Shelley Roth, Andrew Caporaso, Jill DeMatteis.

**Formal analysis:** Shelley Roth, Andrew Caporaso, Jill DeMatteis.

**Investigation:** Shelley Roth, Andrew Caporaso, Jill DeMatteis.

**Methodology:** Shelley Roth, Andrew Caporaso, Jill DeMatteis.

**Supervision:** Jill DeMatteis.

**Visualization:** Andrew Caporaso.

**Writing – original draft:** Shelley Roth, Andrew Caporaso, Jill DeMatteis.

**Writing – review & editing:** Shelley Roth, Andrew Caporaso, Jill DeMatteis.

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
