## [Decision Letter · Decision Letter 0]

14 Dec 2021

PONE-D-21-31646Variables appended to ABS frames: Has their data quality improved?PLOS ONE

Dear Mr. Andrew Caporaso,

Thank you for submitting your manuscript to PLOS ONE. After careful consideration, we feel that it has merit but does not fully meet PLOS ONE’s publication criteria as it currently stands. Therefore, we invite you to submit a revised version of the manuscript that addresses the points raised during the review process, including the comment from Reviewer#2 about "plagiarism".

We look forward to receiving your revised manuscript.

Kind regards,

Gouranga Lal Dasvarma, PhD

Academic Editor

PLOS ONE

Journal Requirements:

2. Commercial affiliation should be listed as competing interest.

Additional Editor Comments (if provided):

Please address all the comments (including that of Reviewr#2 about "plagiarism").

Reviewers' comments:

Reviewer's Responses to Questions

**Comments to the Author**

1. Is the manuscript technically sound, and do the data support the conclusions?

Reviewer #1: Yes

Reviewer #2: Partly

2. Has the statistical analysis been performed appropriately and rigorously? 

Reviewer #1: Yes

Reviewer #2: Yes

3. Have the authors made all data underlying the findings in their manuscript fully available?

Reviewer #1: No

Reviewer #2: No

4. Is the manuscript presented in an intelligible fashion and written in standard English?

Reviewer #1: Yes

Reviewer #2: Yes

5. Review Comments to the Author

Reviewer #1: Question 3 above befuddled me. The paper goes to great length to indicate that the proprietorial vendors of ABS frames do not always fully explain the nature and methods of their data, so it is not likely that this paper could make all underlying data fully available. But it seems that they have made everything available that was within their control.

This is a very useful study as far as it goes. My comments are in the nature of pushing the boundaries further. Being from overseas, I thought the authors might have helped the foreign readers to understand how the ABS might differ between US, UK, Canada, Australia, and European countries. I know that many Asian countries would struggle to attain the levels of coverage of the USPS.

Hispanic and non-Hispanic is a category I know from the US Census, but it wasn't clear why the vendors did not append questions on race.

The matching exercise in Table 2 is useful, but it is not clear how the "bar" of 90% operates for practical decisions. Is 88% that much different to 90% to be notionally pushed aside? How about 80%? It would be useful to have more discussion of the standards you are developing, and the interpretation of the predictivity.

In the concluding paragraph, this sentence stands out: "Future research should continue to monitor and document the availability and accuracy of different types of appended frame data." Availability is the key here. Can you give examples of what other variables should be brought forward? Race, for instance?

Reviewer #2: General:

• This study is useful for researchers who deal with quantitative survey data –

particularly on the survey sampling design, data analysis and the interpretation of

the findings.

• It presents the original research. However, there is a level of plagiarism (based on

the ‘Turn It In’, please see results in an attached file).

• Results reported have not been published elsewhere. Although, there are number of

similar published publications, especially the reference # 4 that the authors would

like to follow up with an improvement of the appended data over time (line 96).

• Mathematics formula and data analysis performed well in an appropriate manner,

but it need references for methods and formulas.

• Conclusions are clearly drawn from the results.

• The data analysis of this manuscript was based on the secondary survey data sets,

and appended data from a vendor. The research meets all applicable standards for

the ethnics of research integrity.

Specific:

• Introduction:

The rationale of this study should be strengthened, and research gaps should be

clearly identified. These would be aligned with objectives and methodology.

• Data and methods:

The authors end up with using three data sets (NHTS, HINTS, ACS), but under in Table

1, the authors provide 2 detail data sets, while the third dataset was not directly use

for data analysis was presented. Why did the authors give the NHES detail? Please

clarify.

• Availability and quality of appended demographic information:

o Please define ‘demographic information’ used in this study. Normally,

demographers define demographic variables as age, sex, ethnicity/race of the

population.

o Fig.1 and Fig.2, please label x-axis and y-axis.

o Lines 144-145: descriptive, please check its consistency with the figures in

Table 2.

• Efficiency of using appended demographic variables for oversampling

o “Oversampling” was not mentioned in the rationale- please add argument

why researcher should examine efficiency of using appended data for

oversampling.

o Table 3: please clarify why ’18-24 present’ was not highlighted in the table,

while its optimum oversampling rate =2.3 (above 2)

2

o Line 214; according to the results, please clarify why the household income is

the less promising variable.

o The American Community Survey (ACS), please clarify why didn’t the authors

mentioned it at the beginning (in the data and methods session).

o Suggest: that the authors add references for each formula, and show a

sample calculation e.g for the first figures in Tables 2 & 3.

• Discussion:

The authors should add work more on the strengthen and limitation of this study,

and how it support or contrast with the existing knowledge that was reviewed in the

introduction. At this stage, there is almost no discussion in this session.

• References:

Only 15 references were mentioned, while references related to methods and

calculation should be added

6. PLOS authors have the option to publish the peer review history of their article (what does this mean?). If published, this will include your full peer review and any attached files.

Reviewer #1: **Yes: **Terence H. Hull

Reviewer #2: No

---

## [Author Response · Author response to Decision Letter 0]

14 Apr 2022

Thank you to the reviewers for their careful reviews and feedback. Please see attached document for responses to each of the points raised about the manuscript.

---

## [Editor Report · Decision Letter 1]

16 May 2022

Variables appended to ABS frames: Has their data quality improved?

PONE-D-21-31646R1

Dear Dr. Andrew Caporaso,

We’re pleased to inform you that your manuscript has been judged scientifically suitable for publication and will be formally accepted for publication once it meets all outstanding technical requirements.

Kind regards,

Gouranga Lal Dasvarma, PhD

Academic Editor

PLOS ONE
---

## [Editor Report · Acceptance letter]

11 Oct 2022

PONE-D-21-31646R1 

Variables appended to ABS frames: Has their data quality improved? 

Dear Dr. Caporaso:

I'm pleased to inform you that your manuscript has been deemed suitable for publication in PLOS ONE. Congratulations! Your manuscript is now with our production department. 

Kind regards, 

on behalf of

Dr. Gouranga Lal Dasvarma 

Academic Editor

PLOS ONE